# Ultrafast unidirectional spin Hall magnetoresistance driven by terahertz light field

Ruslan Salikhov [1] ✉, Igor Ilyakov[1], Anneke Reinold [2], Jan-Christoph Deinert [1], Thales V. A. G. de Oliveira [1], Alexey Ponomaryov [1], Gulloo Lal Prajapati[1], Patrick Pilch [2], Ahmed Ghalgaoui[2], Max Koch [2], Jürgen Fassbender [1,3], Jürgen Lindner[1], Zhe Wang [2] & Sergey Kovalev [2] ✉

The ultrafast control of magnetisation states in magnetically ordered systems poses significant technological challenges yet is vital for the development of memory devices that operate at picosecond timescales or terahertz (THz) frequencies. Despite considerable efforts achieving convenient ultrafast readout of magnetic states remains an area of active investigation. For practical applications, energy-efficient and cost-effective electrical detection is highly desirable. In this context, unidirectional spin-Hall magnetoresistance (USMR) has been proposed as a straightforward two-terminal geometry for the electrical detection of magnetisation states in magnetic heterostructures. In this work, we demonstrate that USMR is effective at THz frequencies, enabling picosecond time readouts initiated by light fields. We observe ultrafast USMR in various ferromagnet/heavy metal thin film heterostructures via THz second-harmonic generation. Our findings, along with temperature-dependent measurements of USMR, reveal a substantial contribution from electron-magnon spin-flip scattering, highlighting the potential for all-electrical detection of THz magnon modes.

Terahertz (THz) spintronics, a rapidly advancing field, holds the potential to extend communication bandwidth to THz frequencies and achieve ultrafast writing and reading of magnetic states by leveraging energy-efficient, compact, and cost-effective electron spin-based components[1–3]. By exploiting femtosecond laser pulses to stimulate spin currents, research has demonstrated that fundamental spintronic phenomena, including the inverse spin-Hall effect (ISHE)[4], spin-transfer torque[5–7] and spin-Seebeck-related effects[8–10] remain active and efficient even in the ultrafast regime. Furthermore, utilising THz light field as an excitation stimulus has enabled the characterisation of spin-pumping[11,12], the anomalous[13,14] and spin-Hall effects[15], spin-orbit torque[16–18], anisotropic magnetoresistance[19,20], as well as giant magnetoresistance (GMR)[21], tunnel[22] and colossal[23] magnetoresistance within a picosecond timeframe. These insights have enabled the

optimisation of spintronic THz emitters, achieving intensities comparable to state-of-the-art high-field THz sources[24–26]. Furthermore, spintronic THz frequency up-conversion and rectification have been realised using ultrathin ferromagnet/heavy metal (FM/HM) heterostructures[27]. While both spintronic approaches hold promise for THz technology, the spin-current generation in these cases is "thermally" driven, meaning that the spin-current density is proportional to the absorbed instantaneous intensity of the optical or THz pulses[27–30].

To expand the functionalities of THz spintronics, it is crucial to explore ultrafast versions of other spintronic effects coherently driven with light fields. Ultrafast spin-Hall magnetoresistance (SMR) has been theoretically investigated to gain insights into spin transport across FM/HM interfaces on a picosecond timescale, showing promise as a

[1]Helmholtz-Zentrum Dresden-Rossendorf, Dresden, Germany. [2]Department of Physics, TU Dortmund University, Dortmund, Germany. [3]Institute of Solid State and Materials Physics, TU Dresden University, Dresden, Germany. ✉e-mail: r.salikhov@hzdr.de; sergey.kovalev@tu-dortmund.de

tool for all-electric spectroscopy of magnon modes[31]. In addition to GMR and SMR, unidirectional spin-Hall magnetoresistance (USMR) has been identified as another spin-current-related phenomenon affecting electrical conductivity in FM/HM layers[32–37]. USMR arises from spin-dependent scattering at the FM/HM interface and, unlike SMR, it is proportional to the magnitude of the electric current. Furthermore, USMR changes sign upon the reversal of FM magnetisation or the direction of the electric current. This results in resistance exhibiting periodic modulation with the current, thereby generating a nonlinear second harmonic signal when alternating currents pass through the HM layer[32–37].

Here we report on the THz fields-driven ultrafast version of the USMR effect in metallic FM/HM systems, which results in THz second harmonic generation (SHG) detected in a far-field configuration. We demonstrate experimentally that the SHG signal is odd under THz currents or magnetisation inversion, confirming the ultrafast USMR origin. Additionally, by analysing the in-plane angular dependence between THz currents and magnetisation direction, we can disentangle the USMR contribution to the SHG signal from the "thermally" driven SHG reported recently[27]. Our results enable the study of spin-dependent scattering at FM/HM interfaces with sub-ps temporal resolution in a non-destructive manner and without requiring any lithographic patterning. Consequently, these studies can be readily extended to topological insulators[38,39], Rashba-type[40–42], and orbital-type[43] unidirectional magnetoresistance, opening avenues for improving the efficiency of spintronic THz converters[44,45]. Furthermore, a sizable ultrafast USMR effect allows for the electrical detection of spin states in antiferromagnets using picosecond-time electric currents[46–49].

## Results

For our studies, we used nonlinear THz time-domain spectroscopy (TDS) in transmission geometry. Spectrally dense narrow-band THz radiation with a centre frequency $\Omega = 0.3$ THz and approximately 10% bandwidth was generated by the superradiant THz undulator source driven by the high-repetition rate linear accelerator at the ELBE facility located at the Helmholtz-Zentrum Dresden-Rossendorf[50]. A pair of wire grid polarizers (WG) was used to control the polarisation of the THz pulses while ensuring that their peak field strength remained at approximately 100 kV/cm. Additionally, a THz quarter-wave plate (QWP) was employed to control the beam's ellipticity, enabling the adjustment of the polarisation from linear to circular as needed. A THz beam with a full width at half maximum spot size of approximately 0.8 mm was focused on the sample. The SHG emitted from the sample at $2\Omega = 0.6$ THz was refocused onto a 2 mm thick $\langle 110 \rangle$ ZnTe single crystal for electro-optic sampling (EOS). To perform EOS, 35 fs laser pulses with a central wavelength of 800 nm were used from an external laser system that was synchronised with the ELBE accelerator[51]. To detect the SHG polarisation, the ZnTe crystallographic orientation was manually adjusted in conjunction with an additional WG placed between the sample and the EOS crystal. Furthermore, THz bandpass filters were employed to diminish the influence of the THz pump beam on the SHG signal. For further details on the THz TDS setup, refer to the Supplementary Information.

We fabricated trilayer Ta(2nm)/Py(3nm)/Pt(2nm) and Ta(2nm)/Co(2nm)/Pt(2nm), as well as bilayer FM(2nm)/HM(3nm) (FM = Py, Co, Ni; HM = Pt, W) heterostructures, where Py represents $Ni_{81}Fe_{19}$, with the thickness of each layer specified in parentheses. All samples were capped with a 10-nm-thick $SiO_x$ layer to prevent any potential oxidation of the top layer. The Ta in the trilayer structures served multiple purposes: i) as an adhesion layer to a 1-mm-thick quartz glass substrate; ii) as a Ta buffer layer to facilitate smoother interfaces between all layers; iii) it has been suggested that the USMR contribution can be enhanced by sandwiching the FM (Py and Co) layer between HM (Ta and Pt) layers, which exhibit opposite signs of the spin Hall

conductivities[34]. In the THz TDS, we employed a permanent magnet to create a magnetic field of approximately 100 mT at the sample position to ensure in-plane magnetic saturation of the Py and Co layers.

The concept of THz SHG in FM/HM systems due to USMR is schematically illustrated in Fig. 1(a). In each half-period of the narrow-band THz field ($\mathbf{E}_{THz}^{\Omega}$), which oscillates at $\Omega = 0.3$ THz and is polarised perpendicular to the FM magnetisation direction, the THz-induced electric currents ($\mathbf{j}_e$) in the HM result in spin currents ($\mathbf{j}_s$) due to the SHE. The direction of these ultrafast spin currents is periodically modulated by the THz field. During each half-cycle, the SHE-induced spin polarisation is parallel (or antiparallel) to the FM magnetisation, leading to an increase (or decrease) in the interface resistivity ($\rho^{\uparrow\uparrow} \neq \rho^{\uparrow\downarrow}$), as shown in Fig. 1(a). The modulation of the interface resistivity at the frequency of $\Omega$ results in SHG, with an amplitude proportional to USMR (see Supplementary Information). This amplitude can be estimated as $\mathbf{E}_{USMR}^{2\Omega} \sim (\rho^{\uparrow\uparrow} - \rho^{\uparrow\downarrow}) \mathbf{E}_{THz}^{\Omega} \sim ([\mathbf{e} \times \mathbf{j}_s], \mathbf{m}) \mathbf{E}_{THz}^{\Omega}$, where $\mathbf{e}$ is the THz pulse polarisation, $[\mathbf{e} \times \mathbf{j}_s]$ represents the accumulated spin polarisation at HM interfaces due to SHE. The electric current $\mathbf{j}_e$ is proportional to $\mathbf{E}_{THz}^{\Omega}$ according to Ohm's law, and $\mathbf{m}$ is the unit vector of the magnetisation in the FM layer. Thus, the USMR-SHG signal ($\mathbf{E}_{USMR}^{2\Omega}$) can be detected when the polarisation of the THz field $\mathbf{E}_{THz}^{\Omega}$ is not collinear with the direction of the magnetisation in the FM layer[32–37].

Besides the ultrafast USMR contribution, another spintronic mechanism at the FM/HM interfaces also triggers the SHG[27]. In this case, SHG is the result of either a THz field-driven ultrafast spin See-beck effect, ultrafast demagnetisation, or a combination of both. Here, the THz excitation causes an instantaneous increase in the electron temperature of the HM and FM layers, which in turn results in the generation of a spin current ($\mathbf{j}_s^{FM}$) at the FM/HM interface. The generated FM spin current (Fig. 1b) is converted into charge current ($\mathbf{j}_{ISHE}$) in the HM layer via the ISHE. Since this current is not determined by the orientation but by the instantaneous intensity of the THz pulse, the ISHE current oscillates at twice the frequency of the THz field, resulting in THz second harmonic generation[27]. This contribution, which we refer to as ISHE-SHG, is independent of the THz pump polarisation angle and is polarised orthogonally to the magnetisation direction of the FM layer.

Given the different origins and geometries of the spintronic SHG contributions at FM/HM interfaces, these can be readily disentangled using two distinct experimental schemes presented in Fig. 1c, d. These schemes differ only in the direction of the magnetisation in the FM layer, which is either vertical (along the $x$-axis, Fig. 1c) or horizontal (along the $y$-axis, Fig. 1(d)). In all our experiments, the EOS and WG were adjusted to sense only the horizontal polarisation components of the SHG signal. Thus, for vertical magnetisation, the SHG from the ISHE ($E_{ISHE}^{2\Omega}$) can be detected regardless of the THz pump polarisation angle, whereas the USMR contribution ($E_{USMR}^{2\Omega}$) shows a maximum at the horizontal polarisation of the THz pump. The symmetry of both components is schematically illustrated in Fig. 1e. It can be recognised that at the horizontal polarisation of the pump beam, both components can be detected, whereas only the $E_{ISHE}^{2\Omega}$ contribution is possible at vertical polarisation. In the case of the horizontally magnetised FM layer (Fig. 1d), the vertically polarised $E_{ISHE}^{2\Omega}$ component cannot be detected in our experimental setup. However, the $E_{USMR}^{2\Omega}$ contribution can be detected at any THz pump polarisation angle that deviates from either the horizontal direction (where the THz field is parallel to $\mathbf{m}$) or the vertical direction (where $E_{USMR}^{2\Omega}$ is filtered by the WG polarisers). The maximum signal is expected at a polarisation angle of $\phi = 45°$, as illustrated in Fig. 1f.

To illustrate that the SHG signal contains two distinct contributions, in Fig. 2a, b we present time-delay scans of the SHG field obtained with horizontal and vertical THz pump polarisations for vertically magnetised (see the scheme in Fig. 1c) Ta(2nm)/Py(3nm)/Pt(2nm) and Ta(2nm)/Co(2nm)/Pt(2nm) samples, respectively. The

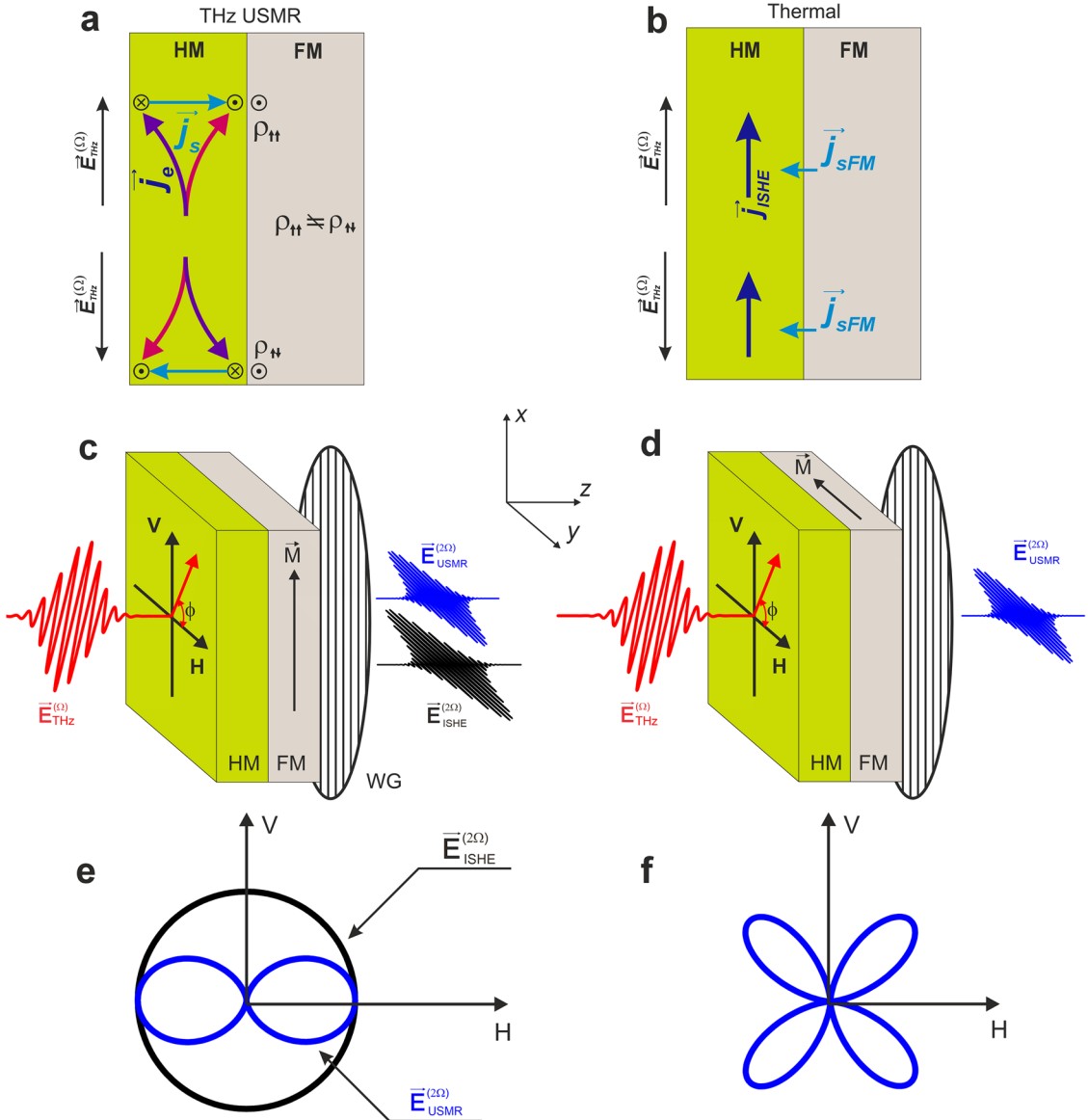

**Fig. 1 | Schematics of spintronic THz frequency conversion components and their detection. a, b** Representation of (**a**) the USMR-SHG, and (**b**) the ISHE-SHG. **a** At each half-cycle of the THz field ($E_{THz}^{\Omega}$) impinging on the FM/HM system, electric currents ($j_e$) in the HM generate spin currents ($j_s$) due to the SHE. This results in spin accumulation ( $\odot$ or $\otimes$, depending on the direction $E_{THz}^{\Omega}$) at the interface with the FM layer, which is in-plane magnetised along the $y$-axis ( $\odot$ in the FM layer denotes its magnetisation direction). Consequently, at each THz field period, the interface resistivity ($\rho$) is modulated, as $\rho^{\uparrow\uparrow} \neq \rho^{\uparrow\downarrow}$ due to the USMR, leading to SHG ($E_{USMR}^{2\Omega}$). **b** In each THz half-cycle, the induced currents in the FM layer lead to its demagnetisation. The generated spin currents ($j_s^{FM}$) due to demagnetisation are converted into electric currents ($j_{ISHE}$), which emit an electromagnetic field. This phenomenon

results in ISHE-SHG ($E_{ISHE}^{2\Omega}$). **c, d** Detection method used to disentangle USMR-SHG and ISHE-SHG contributions. The FM magnetisation (M) is aligned either **c** vertically (along the $x$-axis, V) or **d** horizontally (along the $y$-axis, H). The wire grid (WG) is adjusted to sense only the horizontal component of the SHG signal. In the vertical magnetisation geometry, the $E_{ISHE}^{2\Omega}$ signal will be detected regardless of the polarisation angle of the THz pump pulse. The $E_{USMR}^{2\Omega}$ contribution will be detected only with the horizontal pump, as the USMR-SHG polarisation is collinear with the polarisation of $E_{THz}^{\Omega}$. **e** illustrates the dependence of SHG contributions as a function of THz pump polarisation angle ($\phi$). In the horizontal magnetisation geometry, the $E_{ISHE}^{2\Omega}$ signal is not detected as it is filtered by the WG. Only the $E_{USMR}^{2\Omega}$ contribution can be detected (**f**).

complete angular dependence of the SHG intensity across these angles is presented in Fig. 3(a). As discussed above, under horizontal THz pump polarisation, the SHG signal is anticipated to contain both $E_{ISHE}^{2\Omega}$ and $E_{USMR}^{2\Omega}$ components, whereas vertical polarisation gives rise solely to the $E_{ISHE}^{2\Omega}$ signal (Fig. 1e). The larger signal observed with a horizontally polarised THz field indicates a significant contribution of the USMR in comparison to the ISHE-SHG in both samples. However, both the $E_{ISHE}^{2\Omega}$ and $E_{USMR}^{2\Omega}$ contributions are smaller in the Ta(2nm)/Co(2nm)/Pt(2nm) sample, a point which is discussed further below. In the Ta(2nm)/Py(3nm)/Pt(2nm) sample, the USMR contribution to the ISHE-SHG signal ($E_{USMR}^{2\Omega}/E_{ISHE}^{2\Omega}$) is approximately 60%. Given the

spintronic SHG efficiency $E_{ISHE}^{2\Omega}/E_{THz}^{\Omega} \approx 10^{-4}$ [27] and considering that the absorbed power in the most conductive 2-nm-thin Pt layer is about 10% of the THz power (or 30% of the THz field)[17], we estimated the USMR contribution to the total resistivity ($\rho$) as follows:

$$\frac{(\rho^{\uparrow\uparrow} - \rho^{\downarrow\uparrow})}{\rho} \approx \frac{E_{USMR}^{2\Omega}}{E_{abs}^{\Omega}} \approx \frac{0.6 \cdot E_{ISHE}^{2\Omega}}{0.3 \cdot E_{THz}^{\Omega}} \approx 2 \cdot 10^{-4} = 0.02\%, \quad (1)$$

where $E_{abs}^{\Omega}$ is the absorbed THz pump field in the Pt layer. Considering the peak current density in the Pt layer of $2 \cdot 10^7$ A cm$^{-2}$, the calculated THz USMR for the Ta(2nm)/Py(3nm)/Pt(2nm) sample is five times

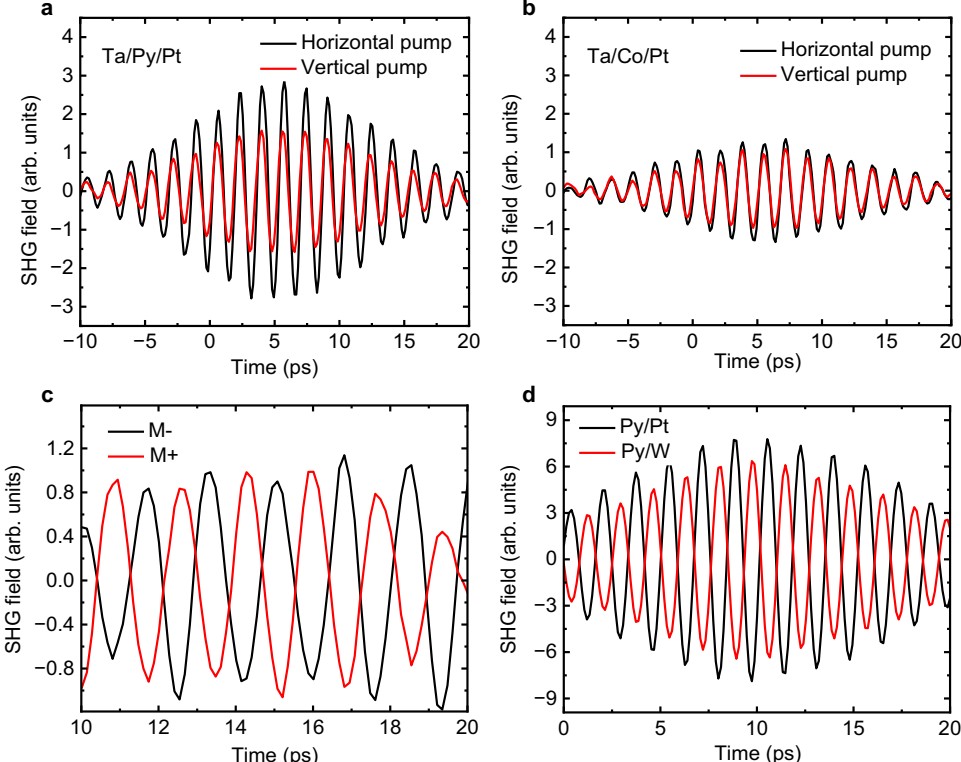

**Fig. 2 | USMR-SHG symmetry. a, b** SHG amplitude in **a** Ta(2nm)/Py(3nm)/Pt(2nm) and **b** Ta(2nm)/Co(2nm)/Pt(2nm) samples, measured in the vertical magnetisation geometry (cf. Fig. 1c) with the THz pump pulse polarised perpendicular (horizontal pump, black curve) and parallel (vertical pump, red curve) to the Py and Co magnetisation direction. The EOS and WG are adjusted to detect only the horizontal polarisation components of the SHG signal. **c** Comparison of SHG signals in the Ta(2nm)/Py(3nm)/Pt (2nm) sample for two opposite magnetisation directions.

Signals were measured in the horizontal magnetisation geometry (Fig. 1d), with the THz pump polarised at $\phi = 45°$ with respect to the magnetisation plane. The 180° phase shift in the signal indicates its odd behaviour under the inversion of the magnetisation direction. **d** Comparison of the SHG signals from Py(2nm)/Pt(3nm) and Py(2nm)/W(3nm) samples measured in the same geometry as in (**c**). The observed 180° phase shift between the signals of the two samples is ascribed to the opposite signs of the spin Hall conductivity in Pt and W.

larger compared to the literature values for the USMR of Co/Pt interfaces[32–37]. However, the observation that the USMR-SHG signal in the Ta(2nm)/Co(2nm)/Pt(2nm) sample is approximately five times smaller compared to the Py layer sample aligns well with the literature[32–37].

To demonstrate that the $E_{USMR}^{2\Omega}$ contribution is odd under magnetisation reversal, and thus satisfies the required symmetry for the USMR[32–37], we isolate this contribution by aligning the Py magnetisation in the horizontal direction and setting the THz pump polarisation angle to $\phi = 45°$, as illustrated in Fig. 1d. The angular dependence of the SHG intensity for this geometry is shown in Fig. 3b. As evident in Fig. 2c, the SHG fields exhibit a 180-degree phase shift for opposite magnetisation directions, confirming the USMR origin of the signal. To further validate this origin, Fig. 2d presents a comparison of signals from two bilayer structures, Py(2nm)/Pt(3nm) and Py(2nm)/W(3nm), measured under identical conditions. The 180-degree phase shift in signals between these samples is due to Pt and W having opposite signs in spin-Hall conductivity, further confirming the USMR origin of the signals[32].

Finally, in Fig. 3a, b, we present the angular dependence of the SHG intensity on THz pump polarisation for vertically magnetised (see the geometry in Fig. 1c) and horizontally magnetised (see the geometry in Fig. 1d) Ta(2nm)/Py(3nm)/Pt(2nm) sample. Both graphs exhibit the expected behaviour, for which we provide fitting considering the symmetry of both, the $E_{ISHE}^{(2\Omega)}$ and $E_{USMR}^{(2\Omega)}$:

$$E_{ISHE}^{2\Omega,\perp} \sim E_0^2, \quad E_{ISHE}^{2\Omega,\parallel} = 0$$
$$E_{USMR}^{2\Omega,\perp} \sim E_0^2 \sin\phi\cos\phi, \quad E_{USMR}^{2\Omega,\parallel} \sim E_0^2 \sin^2\phi \tag{2}$$

where the symbols $\perp$ and $\parallel$ indicate the vertical and horizontal sample magnetisation, respectively. $\phi$ represents the polarisation angle of the THz pump, with $\phi = 0$ indicating horizontal polarisation (for more details, see the Supplementary Information). From the fit, we verified the ratio $E_{USMR}^{2\Omega}/E_{ISHE}^{2\Omega} = 0.6$.

We further conducted a detailed study on the impact of THz pump ellipticity on spintronic SHG in the Ta(2nm)/Py(3nm)/Pt(2nm) sample. Similar to THz third harmonic generation in various materials[15,52,53], the ISHE-SHG exhibits a significant decrease with increasing pump beam ellipticity, eventually reaching zero values under a circularly polarised THz pump[27]. This phenomenon is attributed to the reduction of THz instantaneous intensity modulation with beam ellipticity resulting in a decrease in the amplitude of the spin current density oscillating at the THz SHG frequency from the FM layer[27,53]. On the other hand, the USMR-SHG arises from a mechanism with different symmetry, where the electron spin accumulated by the SHE in the HM is projected onto the FM layer's magnetisation direction[32]. Consequently, the USMR-SHG contribution persists even when the THz pump is circularly polarised. In the Supplementary Information, we provide details on the dependence of both spintronic SHG contributions on the THz pump beam ellipticity, which can be expressed as follows:

$$E_{ISHE}^{2\Omega,\perp} \sim E_0^2 \frac{1-\sin^2 2\chi}{1+\sin^2 2\chi}, \quad E_{ISHE}^{2\Omega,\parallel} = 0,$$
$$E_{USMR}^{2\Omega,\perp} \sim E_0^2 \frac{\sin 2\chi}{1+\sin^2 2\chi}, \quad E_{USMR}^{2\Omega,\parallel} \sim E_0^2 \frac{1}{2(1+\sin^2 2\chi)}, \tag{3}$$

where $\chi$ represents the orientation of the QWP used to induce the ellipticity of the THz pump pulse. The dependence of the SHG intensity

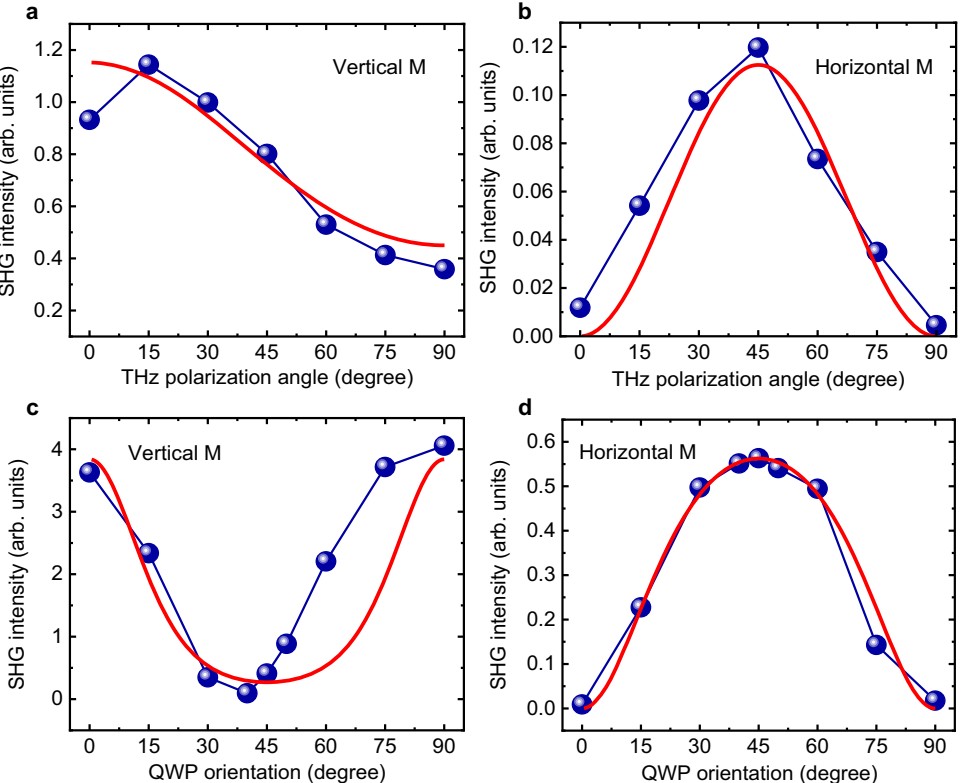

**Fig. 3 | Dependence on THz pump polarisation angle and ellipticity. a, b** SHG intensity in the Ta(2nm)/Py(3nm)/Pt(2nm) sample as a function of the THz pump polarisation angle ($\phi$), measured in (**a**) vertical magnetisation geometry (Fig. 1(c)) and **b** horizontal magnetisation geometry (Fig. 1d). $\phi = 0$ corresponds to the horizontal polarisation. **c, d** The dependence of SHG intensity on THz pump beam ellipticity, measured in (**c**) vertical magnetisation geometry and **d** horizontal

magnetisation geometry. The pump beam's ellipticity was adjusted using a THz quarter-wave plate (QWP) by varying its orientation angle $\chi$ from 0° to 90°. The beam polarisation is circular at $\chi = 45°$. The red lines in all graphs depict simulation results that consider the symmetry of $E^{2\Omega}_{ISHE}$ and $E^{2\Omega}_{USMR}$ signals and are fitted to the experimental data points. In all graphs, the statistical error of the experimental data does not exceed the symbol size.

on the QWP angle for vertical and horizontal magnetisation orientations is presented in Fig. 3c, d, respectively, for both experimental (blue circles) and simulated (red line) data. Note that at the QWP angles $\chi = 0°$ and 90°, the THz pump beam is identically linearly polarised, and it becomes elliptical at any other angles, except at 45°, where the beam becomes circularly polarised. A good agreement between analytical calculations and the experimental results confirms the symmetries of both spintronic contributions to the THz SHG signal detected in our experimental geometries. Similar to the angular dependence on the THz polarisation shown in Fig. 3a, b, we observed an identical contribution of the USMR-SHG signal relative to the ISHE-SHG, with $E^{2\Omega}_{USMR}/E^{2\Omega}_{ISHE} = 0.6$.

The discrepancies between the data points and the simulation curves presented in Fig. 3a–d are attributed to THz pump-pulse instabilities caused by irregular power losses from the accelerator source. Additionally, imperfections in the THz quarter-wave plate used for ellipticity dependence may lead to interference between the ISHE-SHG and USMR-SHG signals, resulting in deviations from the proposed model. We note that both ISHE and USMR contributions are present in the data shown in Fig. 3c, while Fig. 3d displays only the USMR contribution, where the discrepancy between the model and the data is minimal.

## Discussion

The dependencies on THz-field polarisation angle and THz beam ellipticity in Fig. 3 revealed an additional contribution to the thermally driven SHG that exhibits the symmetry characteristic of USMR. Similar to the ISHE-SHG[27], the USMR-SHG signal displays quadratic scaling with the THz pump field amplitude, as shown in the Supplementary

Information. The spintronic origin of the effect is confirmed by the fact that this contribution is odd under the inversion of FM magnetisation and the sign of the spin Hall conductivity (Fig. 2c, d). The microscopic origin of the ultrafast USMR is identical to that reported in magneto-transport measurements[32–37], as the typical electron spin-flip time in HMs is less than 30 fs[54,55], which is shorter than the THz half-cycle time of 1.7 ps at a frequency of 0.3 THz. Accordingly, one expects a similar USMR contribution to the total resistance (USMR ratio) in metallic FM/HM systems, typically ranging between 0.002% and 0.005%[32–37]. Based on the THz transmission data of the studied materials in ref. 17, we roughly estimated a similar ratio for the Ta(2nm)/Co(2nm)/Pt(2nm) sample. However, the USMR ratio for the Ta(2nm)/Py(3nm)/Pt(2nm) sample is approximately five times larger.

In Fig. 4a, we present a comparison of the $E^{2\Omega}_{USMR}$ values measured in the horizontal magnetisation geometry at a THz pump polarisation angle of $\phi = 45°$, ensuring that only the USMR-SHG signal was detected for all studied systems (Fig.1d, f). The data are calculated as the square root of the spectral integrated intensity to ensure the accuracy of the compared amplitudes. We first notice that for all FM layers (Py, Co, Ni) in the FM/HM bilayers, the signal is systematically smaller when the HM is W compared to Pt. This is expected, as the spin Hall conductivity in our magnetron sputter-deposited W films is lower than that of Pt[15,56]. Secondly, there is a significant dependence of the signal amplitude on the FM material in the FM/HM bilayers, regardless of whether the HM is Pt or W. The significant variation in spintronic characteristics with different FM layers is attributed to numerous parameters at the FM/HM interface, which strongly influence both the spin-to-charge inter-conversion (the origin of the ISHE-SHG, $E^{2\Omega}_{ISHE}$) and the charge-to-spin interconversion (the origin of the USMR-SHG, $E^{2\Omega}_{USMR}$). Besides the spin

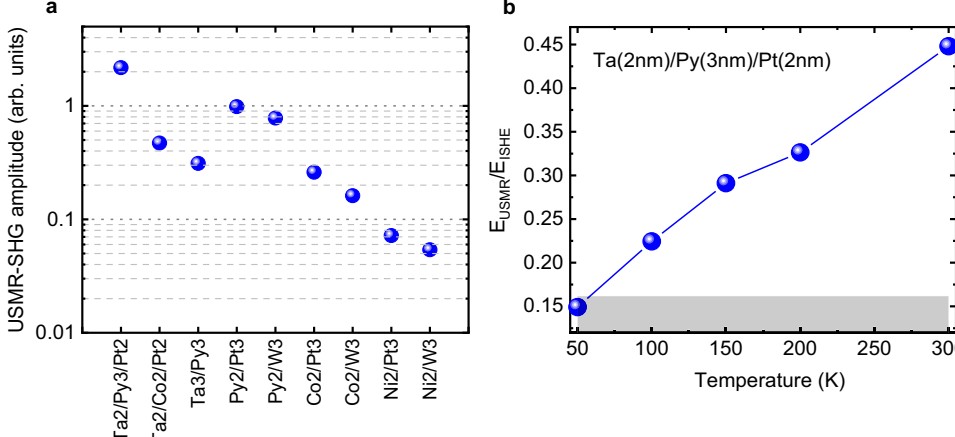

**Fig. 4 | Material and temperature dependence of USMR-SHG signals. a** A summary of USMR-SHG amplitudes for samples with different material compositions and thicknesses. The thickness (in nm) of each individual layer is indicated on the horizontal axis label. **b** Temperature dependence of the USMR-SHG signal, represented as the $E^{2\Omega}_{USMR}/E^{2\Omega}_{ISHE}$ ratio, to exclude experimental artifacts related to the drift of the overall signal due to thermal contractions. The grey shaded area indicates the noise floor limit. All USMR-SHG components were measured in the horizontal magnetisation geometry (Fig. 1(d)) at an angle of $\phi = 45°$. The thermally driven SHG signal was measured in the vertical magnetisation geometry (Fig. 1c) with a vertically polarised THz pump pulse.

Hall conductivity and spin diffusion length in an HM, these parameters include the FM spin-asymmetry (spin polarisation), its conductivity, and its spin diffusion length. Additionally, the interface spin-dependent resistance (SDR), as well as the spin memory loss (SML) parameter, which accounts for interfacial spin-flip scattering and depends on the interface materials, morphology (lattice mismatch, material interdiffusion) and temperature, are also crucial[57,58]. A significant difference in the aforementioned parameters might explain the small USMR-SHG signal observed in FM = Ni bilayers. Notably, the ISHE-SHG contribution is also the smallest in the Ni samples.

Although the stronger signal from Py/Pt bilayers compared to Co/Pt is consistent with theoretical predictions that account for interfacial SDR and SML[58,59], the observed fourfold increase in USMR-SHG amplitudes when interfacing Pt with Py (Fig. 4a) is too large for the proposed models[58,59]. Furthermore, experimental studies using spin-torque ferromagnetic resonance or HM wires connected to transverse FM electrodes have reported much stronger signals for Co/Pt interfaces compared to Py/Pt[60,61]. Even with the beneficial effects of the Ta buffer layer on the Ta(2nm)/Co(2nm)/Pt(2nm) sample, enhancing the amplitude through additional USMR contributions from Ta, improved interface morphology, or modifications in the Co-layer crystallographic phase[62], the signal from the "optimised" Co sample remains smaller in comparison to all Py/Pt samples.

While one might suggest that increasing the thickness of the Co layer could help elucidate the USMR discrepancy between Co and Py samples, the maximum USMR in the thickness dependence is observed at a 3 nm Co layer[36]. However, the USMR is only 50% larger compared to the thinner Co thicknesses[36]. We propose that the significant enhancement of USMR in Py samples stems from an increased spin-flip contribution due to electron-magnon scattering[33,63,64]. Depending on the spin direction accumulated by the SHE at the FM/HM interface during each THz half-cycle, electron-magnon scattering leads to the creation or annihilation of magnons. Consequently, this process causes a decrease or increase in the electron spin-flip time within the FM layer, resulting in periodic modulations in the longitudinal resistance. A large USMR ratio in Py/Pt bilayers was reported in ref. 64 and was primarily attributed to the dominant magnonic contribution. Py and Co exhibit distinct electron-magnon scattering characteristics ("interband" in Py compared to "intraband" in Co) and possess significantly different spin diffusion lengths and spin-flip times[65,66], resulting in different magnonic impacts on USMR in each material. While the direct verification of the THz magnon contribution to the ultrafast USMR

requires further investigation, such as studying the impact of magnetic resonance excitation on USMR properties[31], exploring the temperature dependence could provide additional insights[33,43].

The ISHE-SHG exhibits weak temperature dependence, showing a 25% decrease when the temperature is reduced from 300 K to 50 K[27]. This behaviour is attributed to the decrease in resistivity of the Pt layer, which attenuates the amplitude of the THz SHG emission since the intrinsic spin Hall resistivity scales quadratically with the resistivity of Pt[27,67]. In Fig. 4b, the temperature dependence of the $E^{2\Omega}_{USMR}/E^{2\Omega}_{ISHE}$ ratio for the Ta(2nm)/Py(3nm)/Pt(2nm) sample is shown. The figure demonstrates a nearly threefold stronger decrease in the USMR-SHG signal compared to the ISHE-SHG signal when the temperature is reduced from 300 K to 50 K. This observation emphasises an additional contribution to the SDR and SML in the USMR-SHG process with a much more pronounced temperature dependency. Avci et al.[33] have previously shown the strong temperature dependence of the electron-magnon scattering contribution to the USMR, supporting our interpretation for the fivefold increase in the USMR ratio in the Ta(2nm)/Py(3nm)/Pt(2nm) sample compared to the Ta(2nm)/Co(2nm)/Pt(2nm) system and the existing literature data.

In conclusion, the substantial ultrafast USMR effect[68], along with thermally driven THz excitations, could play a key role in optimising spintronic THz frequency multiplication and rectification devices. The THz light field-driven ultrafast USMR effect facilitates the detection of magnetisation states through the direct measurement of the electrical field from the SHG signal. This detection method allows for operation without a laser system, instead utilising bolometer-type detectors or THz diodes. Furthermore, the electrical detection of the rectified signal associated with THz SHG permits rapid readout on a sub-nanosecond timescale, leveraging the modulation of THz radiation. The substantial impact of electron-magnon scattering on the USMR effect enhances the electrical detection of THz magnons, providing an order of magnitude increase in sensitivity compared to spin-pumping detection. This advancement represents a significant step towards spintronic and magnonic technologies operating at THz frequencies.

## Methods

All samples listed in Fig. 4a were fabricated at room temperature using d.c. magnetron sputter deposition in a $3 \times 10^{-3}$ mbar Ar atmosphere within an ultrahigh-vacuum BESTEC system with a base pressure of $5 \times 10^{-9}$ mbar. The heterostructures were all grown on double-side polished quartz glass (SiO$_2$) substrates and to prevent surface

degradation from oxidation, each sample was capped with a 10 nm $SiO_x$ layer using r.f. sputter deposition. In order to ensure uniform growth of the metallic layers, the sample holder was rotated at 30 rpm. The thickness of each layer was controlled by the deposition time.

## Data availability

All data are available in the main text or the supplementary information. The source data are available at https://doi.org/10.14278/rodare.3116. All other data that support the finding of this work are available from the corresponding authors upon request.

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

## Acknowledgements

The authors express their gratitude to Thomas Naumann and Jakob Heinze for their technical support with the sample deposition tools. Parts of this research were carried out at ELBE at the Helmholtz-Zentrum Dresden-Rossendorf e.V., a member of the Helmholtz Association. Z.W. acknowledges support from the European Research Council (ERC) under the Horizon 2020 research and innovation programme, grant agreement no. 950560, and partial support from MERCUR (Mercator Research Centre Ruhr) via Project No. Ko-2021-0027.

## Author contributions

S.K., R.S. initiated and designed the project. R.S. handled the design and fabrication of the samples. S.K., I.I., R.S., A.R., T.V.A.G.O., A.P., G.L.P., and J.-C.D. carried out the experiment at TELBE facility. S.K., A.R. carried out the laser-based experiment. S.K., J.F., J.L., and Z.W. acquire the funding for this work. S.K., R.S. wrote the manuscript with input from I.I., A.R., J.-C.D., T.V.A.G.O., A.P., G.L.P., P.P., A.G., M.K., J.F., J.L., Z.W. All authors provided critical feedback and helped shape the research, analysis and manuscript.

## Funding

## Competing interests

The authors declare no competing interests.
