## [Transparent Peer Review file · Nature Communications]

Ultrafast unidirectional spin Hall magnetoresistance driven by terahertz light field

Corresponding Author: Dr Sergey Kovalev

Version 0:

Reviewer comments:

Reviewer #1

(Remarks to the Author)

The manuscript presents a highly impactful study on unidirectional spin-Hall magnetoresistance (USMR), demonstrating its activity at THz frequencies and highlighting its potential for ultrafast magnetic state detection. The authors' successful detection of ultrafast USMR in various ferromagnet/heavy metal thin film heterostructures through THz second harmonic generation represents a significant advancement in the field of spintronics and THz technologies. The experiments are well-designed, the results are compelling, and the manuscript is written with clarity. I recommend the publication of this manuscript in Nature Communications. Even so, I would like the authors to consider the following point before it can be accepted for publication:

The authors state that USMR can perform ultrafast readout of magnetic states and enables electrical detection. When compared to magneto-optical effects, the manuscript points out that a drawback of magneto-optical methods is their reliance on laser sources and optical components. However, USMR's ultrafast detection in this work also involves the use of laser sources and optical components. I find this somewhat confusing—could you please clarify?

Reviewer #2

(Remarks to the Author)

The authors detected the ultrafast unidirectional spin-Hall magnetoresistance in different types of ferromagnet/heavy metal thin film heterostructures through THz second harmonic generation. And also differentiate the USMR contribution to the THz SHG signal from the ISHE one. These studies provide a method to improve the efficiency of THz frequency converters, which may attract interest of THz spintronics community. The manuscript may be published in Nature Communications, if the authors make the following issues clear.

1. The reasons for the generation of THz SHG signal by USMR are not clear. On page 4, the authors only claimed that "The modulation of the interface resistivity at the frequency of Ω results in SHG". However, it is not well understood the origin. The authors should make a detail explanation.
2. As shown in Fig 1, each half-cycle of THz pulse has reverse electric field and leads to different interface resistivity. If the THz electric field has asymmetric time distribution, should the THz SHG signal increase?
3. The intensity of THz electric field incident on the samples was not given. And how does the THz electric field affect the SHG signal?
4. In Fig3, how to change the THz pump polarization angle? Keep the THz beam unchanged, and rotate the sample and magnet?
5. In Fig 3a, why is the SHG signal with 0 degree of THz polarization angle smaller than that of 15 degree?
6. In Fig 3c, obvious discrepancies can be seen between the experimental data and the fitted data. The authors should make a reasonable explanation. On the other hand, I think the experimental values of the THz SHG signal should be very small. The error bars should be added.
7. On page 7, the authors claimed that the interface morphology may affect the signal. they should add some related characterization.
8. On page 3, the authors wrote "1-nm-thick quartz glass substrate". The thickness of the substrate is only 1 nm?
9. In the abstract, the authors claimed an electrical detection is preferred. How to understand the relation between the THz SHG measurement and the electrical detection?

Reviewer #3

(Remarks to the Author)

The manuscript presents some very interesting results on THz second harmonic generation (SHG) in spintronic structures, and will certainly be of interest to researchers working in the field. The work goes beyond some of the authors earlier work on spintronic terahertz frequency conversion [Nat. Commun. 14, 7010 (2023)], here focusing on separating out a contribution to the SHG signal which is suggested to relate to unidirectional spin-Hall magnetoresistance. The manuscript is well-written, clear in describing the work undertaken, and justifies the conclusions reached.

I assume the authors statement of using a 1-nm-thick quartz glass substrate is a simple typographical error and the thickness is in fact much larger than 1 nm. I would of liked to of seen a little more detail in the methods section to help others recreate the work. For example, giving the physical dimensions of the samples and the permanent magnet used. It would of also been interesting to have measured the full polarization of the THz emitted, prior to detecting only a linearly polarized component, and the authors clearly had the experimental setup for performing such a measurement. I assume that the x- and y-components of the emitted THz field were not recorded due to the limited time available on the TELBE facility? If so, it would be worthwhile making this clear to the reader.

Version 1:

Reviewer comments:

Reviewer #1

(Remarks to the Author)

This paper can be accepted by Nature Communications.

Reviewer #2

(Remarks to the Author)

The authors have made most of my questions clear and also revised the manuscript. So, I can recommend the publication of this manuscript in Nature Communications.

We sincerely thank the Reviewers for their positive feedback and for expressing interest in our work. Below, we provide a detailed point-by-point response to the comments raised. The new text in the revised manuscript is highlighted in magenta to address the reviewers' concerns. Additionally, the Supplementary Information section has been expanded to include further details based on their remarks. We believe that these revisions enhance the manuscript and ensure it meets all the publication criteria for Nature Communications.

Reviewer #1:

The manuscript presents a highly impactful study on unidirectional spin-Hall magnetoresistance (USMR), demonstrating its activity at THz frequencies and highlighting its potential for ultrafast magnetic state detection. The authors' successful detection of ultrafast USMR in various ferromagnet/heavy metal thin film heterostructures through THz second harmonic generation represents a significant advancement in the field of spintronics and THz technologies. The experiments are well-designed, the results are compelling, and the manuscript is written with clarity. I recommend the publication of this manuscript in Nature Communications. Even so, I would like the authors to consider the following point before it can be accepted for publication: The authors state that USMR can perform ultrafast readout of magnetic states and enables electrical detection. When compared to magneto-optical effects, the manuscript points out that a drawback of magneto-optical methods is their reliance on laser sources and optical components. However, USMR's ultrafast detection in this work also involves the use of laser sources and optical components. I find this somewhat confusing—could you please clarify?

Reply:

We sincerely thank the Reviewer for the positive feedback on our work and for the recommendation of publication in Nature Communications.

To clarify, the time-domain THz spectroscopy utilized in our study employs laser pulses from an external source that is synchronized with the linear accelerator for electro-optical sampling. We understand that this may lead to confusion when comparing our experiment with magneto-optical methods, which also rely on laser sources and optical components. To address this, we have revised the abstract to omit discussions of laser sources in the context of magneto-optical detection of magnetization states, thereby minimizing potential confusion.

Fundamentally, our work demonstrates that the ultrafast USMR effect enables the detection of magnetization states through the direct measurement of the electrical field generated by the THz second-harmonic generation (SHG) signal. Notably, this detection method can be performed without a laser system by using direct THz pulse energy measurements with bolometers or THz diodes.

For practical applications, the rectified signal associated with SHG is particularly significant. This signal can be detected electrically as either a DC voltage resulting from continuous THz radiation impinging on the sample, or as a rectified signal in the GHz frequency range when utilizing THz

pulses in our experiments. This detection mechanism is feasible with oscilloscopes that provide sub-nanosecond time resolution. Currently, we are in the process of designing this detection mechanism at our experimental end station, as a part of the user facility at TELBE.

The perspectives of the ultrafast USMR effect for rapid all-electrical detection of magnetic states are discussed in the conclusion section of the revised manuscript.

During the revision process, we became aware of a recent study that presents a significant advancement in the USMR value, resulting in increased output USMR voltage for magnetoresistive random-access memory (MRAM) technology. We have added this new reference as Ref. 68 in the revised manuscript.

We hope this clarification addresses the Reviewer's concern and enhances the discussions in our manuscript.

Reviewer #2:

The authors detected the ultrafast unidirectional spin-Hall magnetoresistance in different types of ferromagnet/heavy metal thin film heterostructures through THz second harmonic generation. And also differentiate the USMR contribution to the THz SHG signal from the ISHE one. These studies provide a method to improve the efficiency of THz frequency converters, which may attract interest of THz spintronics community. The manuscript may be published in Nature Communications, if the authors make the following issues clear.

Reply:

We thank the Reviewer for considering our manuscript suitable for publication in Nature Communications and for his/her constructive comments, which have helped to improve the manuscript. Below, we provide a point-by-point response to the Reviewer's comments.

Comment 1:

The reasons for the generation of THz SHG signal by USMR are not clear. On page 4, the authors only claimed that "The modulation of the interface resistivity at the frequency of Ω results in SHG". However, it is not well understood the origin. The authors should make a detail explanation.

Reply 1:

The origin of harmonic generation as a nonlinear phenomenon arises from the dependence of the sample's electrical conductivity on the THz electric field. By considering the linear contribution to electrical conductivity, we express it as $\sigma(E_{\text{THz}}) = \sigma_0 + \sigma_1 E_{\text{THz}}$, where $\sigma_1 E_{\text{THz}}$ term corresponds to the USMR contribution. This allows us to derive the conditions for SHG presented in the revised Supplementary Information. In that section, we have included a more detailed explanation of the origin of SHG in the context of the propagated THz light.

Comment 2:

As shown in Fig 1, each half-cycle of THz pulse has reverse electric field and leads to different interface resistivity. If the THz electric field has asymmetric time distribution, should the THz SHG signal increase?

Reply 2:

This is indeed an interesting question. When the THz pulse exhibits an asymmetric temporal shape, it leads to the presence of even harmonics in its spectrum. In this case, the THz SHG resulting from the USMR can interfere either constructively or destructively with the initial radiation at the SHG frequencies, depending on the phase relationships between the fundamental radiation and the radiation at the SHG frequencies originating from the asymmetric THz pulse profile. Constructive interference results in an enhancement of the THz SHG signal. However, we would like to reserve a detailed investigation of these effects for future studies, as our current focus is solely on the initial mechanisms underlying the ultrafast USMR. For this reason, we utilized only narrowband THz pulses containing fundamental frequency content, ensuring that the temporal profile of the pulses is symmetric.

Comment 3:

The intensity of THz electric field incident on the samples was not given. And how does the THz electric field affect the SHG signal?

Reply 3:

In our experiment, the peak field strength of the incident THz pulse with linear polarization was set at approximately 100 kV/cm. The amplitude of the USMR-SHG signal exhibits quadratic scaling with the THz pump field amplitude, which is the expected behavior for second-harmonic generation.

In the revised manuscript, we have included the peak field strength of the THz pulse in the experimental section. Additionally, the quadratic relationship between the USMR-SHG signal and the incident THz field is now detailed in the Supplementary Information.

Comment 4:

In Fig3, how to change the THz pump polarization angle? Keep the THz beam unchanged, and rotate the sample and magnet?

Reply 4:

To change the THz pump polarization angles while maintaining identical field strengths, we employed two wire grid polarizers WG_2 and WG_3 , as illustrated in Supplementary Information Fig. 1. These polarizers were mounted on computer-controlled motorized rotational stages, allowing precise adjustments. The superradiant THz source generates a THz field with a vertical (as defined

in Fig. 1) polarization. Using WG₃, we set the polarization angle of the THz pulse incident on the sample surface. Meanwhile, WG₂ was utilized to regulate the amplitude of the THz pump, ensuring it remained constant across all polarization angles, which are used in the angular dependencies shown in Fig. 3a and 3b. Calibration of both angles and the THz pump field strength was conducted prior to the experimental runs. All these details are now provided in the revised Supplementary Information.

Comment 5:

In Fig 3a, why is the SHG signal with 0 degree of THz polarization angle smaller than that of 15 degree?

Reply 5:

The reduced intensity of the SHG signal at a zero pump polarization angle is attributed to a 10% THz power loss from the accelerator source. The data acquisition and fine alignments took approximately 8 hours for the THz polarization angular dependence shown in Fig. 3a. Due to limited shifts and a tight work schedule, we were unable to repeat the angular dependence measurements. We note that this data point drop does not affect the conclusions drawn from the THz polarization angle and elasticity dependencies presented in Fig. 3a–3d.

Comment 6:

In Fig 3c, obvious discrepancies can be seen between the experimental data and the fitted data. The authors should make a reasonable explanation. On the other hand, I think the experimental values of the THz SHG signal should be very small. The error bars should be added.

Reply 6:

To achieve better statistics, an improved signal-to-noise ratio of the SHG signals, and to account for potential power drift during time delay scans while minimizing the impact of parasitic fundamental radiation, we performed four delay scans per sample magnetization with opposite directions. We then took the difference between signals for opposing magnetization directions. Consequently, the statistical error for each data point does not exceed the size of the symbols for all data presented in Fig. 3a–3d.

We attribute the discrepancy between the model and the recorded data in Fig. 3c to a combination of THz power instability (as mentioned in our Reply 5 and relevant for all data in Fig. 3a–3d) and imperfections in the THz quarter-wave plate (QWP) used for ellipticity dependence. If the THz QWP has a phase retardation of 90 degrees, it not only generates an elliptical beam but also tilts the pulse polarization. This can lead to additional interference between the THz SHG contributions from USMR and ISHE, resulting in deviations from the model. We note that both ISHE and USMR contributions are present in the ellipticity dependence shown in Fig. 3c. In contrast, Fig. 3d displays only the USMR contribution, where the discrepancy between the model and the data is minimal.

The details addressing comments 5 and 6 from the Reviewer have been incorporated into the revised manuscript, both in the main text and in the experimental section of the Supplementary Information.

Comment 7:

On page 7, the authors claimed that the interface morphology may affect the signal. they should add some related characterization.

Reply 7:

We have retracted this speculation in the revised manuscript. To investigate the influence of interface morphology, additional systematic experiments should be designed, such as varying the morphology of the layers through adjustments in Ar pressure, deposition rates, substrate temperature, or post-deposition annealing temperature, while maintaining consistent sample composition and layer thickness. Analysing the morphology of these systems alongside THz transmissivity could provide insights into their effects on USMR-SHG characteristics. This endeavor would constitute a separate project for future investigation.

Comment 8:

On page 3, the authors wrote “1-nm-thick quartz glass substrate”. The thickness of the substrate is only 1 nm?

Reply 8:

We thank the Reviewer for highlighting this typo! The substrates used were 1 mm thick.

Comment 9:

In the abstract, the authors claimed an electrical detection is preferred. How to understand the relation between the THz SHG measurement and the electrical detection?

Reply 9:

This comment from the Reviewer is closely related to the previous comment from the first Reviewer.

Due to the THz USMR effect, the magnetization state of the FM layer is encoded in the parameters of the THz SHG signal. The amplitude of the SHG electric field corresponds to the angle between the magnetization and the polarization of the incident THz radiation, while the SHG phase indicates the direction of the magnetization. Consequently, the ability to detect the SHG field without relying on laser femtosecond pulses facilitates the reading of magnetization states. These detection systems are rapidly advancing, utilizing either bolometer-type detectors or THz diodes.

Additionally, the electrical detection of the rectified signal associated with THz SHG is of particular interest. This method can enable rapid readout within a sub-nanosecond timescale by exploiting

the modulation of THz radiation, potentially allowing for the determination of the Néel vector direction in antiferromagnets (Ref. 46).

The perspectives of the ultrafast USMR effect for rapid all-electrical detection of magnetic states we now discuss in the conclusion section of the revised manuscript.

Reviewer #3:

The manuscript presents some very interesting results on THz second harmonic generation (SHG) in spintronic structures, and will certainly be of interest to researchers working in the field. The work goes beyond some of the authors earlier work on spintronic terahertz frequency conversion [Nat. Commun. 14, 7010 (2023)], here focusing on separating out a contribution to the SHG signal which is suggested to relate to unidirectional spin-Hall magnetoresistance. The manuscript is well-written, clear in describing the work undertaken, and justifies the conclusions reached.

I assume the authors statement of using a 1-nm-thick quartz glass substrate is a simple typographical error and the thickness is in fact much larger than 1 nm. I would of liked to of seen a little more detail in the methods section to help others recreate the work. For example, giving the physical dimensions of the samples and the permanent magnet used. It would of also been interesting to have measured the full polarization of the THz emitted, prior to detecting only a linearly polarized component, and the authors clearly had the experimental setup for performing such a measurement. I assume that the x- and y-components of the emitted THz field were not recorded due to the limited time available on the TELBE facility? If so, it would be worthwhile making this clear to the reader.

Reply:

We sincerely thank the Reviewer for the positive feedback and careful review of our work. We have corrected the typo regarding the substrate thickness; indeed, we used 1 mm thick substrates.

In response to the Reviewer's suggestions and comments from the second Reviewer, we have provided a more detailed description of our experiment in the revised Supplementary Information. This includes information on the dimensions of the samples and magnet, the maximum THz peak field strength focused on the sample surface, the data acquisition procedure, and the principle of adjusting the THz pump polarization angle.

We agree with the referee that it would be good to measure the full polarisation of the emitted radiation, but unfortunately this is very time consuming, and we cannot do it given the limited time available at the user facility. This is because we do the measurements in the time domain. A full polarisation reading of the emitted radiation as a function of the polarisation state of the incident pump pulse and for two different magnetisation directions of the sample will require

several days of measurements. We have addressed this issue in the experimental section of the Supplementary Information.

We hope we have clarified the raised points, and the revised manuscript is suitable for publication in Nature Communications.